# A New Hybrid Algorithm Based on Improved MODE and PF Neighborhood Search for Scheduling Task Graphs in Heterogeneous Distributed Systems

**Nasser Lotfi** [1],* **and Mazyar Ghadiri Nejad** [2]

1   Computer Engineering Department, Cyprus International University, Nicosia 99258, TRNC, Turkey
2   Industrial Engineering Department, Cyprus International University, Nicosia 99258, TRNC, Turkey; mnejad@ciu.edu.tr
*   Correspondence: nlotfi@ciu.edu.tr

**Abstract:** Multi-objective task graph scheduling is a well-known NP-hard problem that plays a significant role in heterogeneous distributed systems. The solution to the problem is expected to optimize all scheduling objectives. Pretty large state-of-the-art algorithms exist in the literature that mostly apply different metaheuristics for solving the problem. This study proposes a new hybrid algorithm comprising an improved multi-objective differential evolution algorithm (DE) and Pareto-front neighborhood search to solve the problem. The novelty of the proposed hybrid method is achieved by improving DE and hybridizing it with the neighborhood search method. The proposed method improves the performance of differential evolution by applying appropriate solution representation as well as effective selection, crossover, and mutation operators. Likewise, the neighborhood search algorithm is applied to improve the extracted Pareto-front and speed up the evolution process. The effectiveness and performance of the developed method are assessed over well-known test problems collected from the related literature. Meanwhile, the values of spacing and hyper-volume metrics are calculated. Moreover, the Wilcoxon signed method is applied to carry out pairwise statistical tests over the obtained results. The obtained results for the makespan, reliability, and flow-time of 50, 18, and 41, respectively, by the proposed hybrid algorithm in the study confirmed that the developed algorithm outperforms all proposed methods considering the performance and quality of objective values.

**Keywords:** evolutionary computation; optimization; processor scheduling; pareto optimization; heuristic algorithms

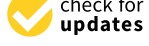



## 1. Introduction

Multi-objective optimization problems comprise some contradicting objectives to be optimized simultaneously when increasing one of them is the cause of decreasing the others [1–3]. To deal with this type of problem, multi-objective optimization algorithms (metaheuristics) are widely applied by researchers, according to recent studies in the literature [4,5]. Multi-objective metaheuristics attempt to find a set of solutions for balancing the trade-off between the problem objectives. Therefore, the goal of metaheuristics is to extract a set of non-dominated solutions, Pareto-front, that optimize all objectives of the problem [6,7]. A non-dominated solution has at least one better objective and no worse objective than all other solutions [2,6]. A sample Pareto-front for minimization of a bi-objective problem is indicated in Figure 1. The represented Pareto-front comprises seven non-dominated and nine dominated solutions out of a total of 16 solutions [8].

Multi-objective task graph scheduling is an NP-hard problem that plays a significant role in the heterogeneous distributed systems. In a task graph scheduling problem, the goal is to distribute all tasks of an acyclic graph (parallel program) over the processors in such a way that all the objective functions are optimized. The solution to the problem

is expected to optimize all the scheduling objectives, such as flow time, reliability, etc., simultaneously [9,10]. A detailed problem definition is presented in Section 2.

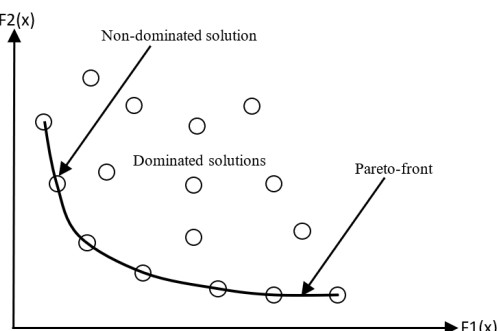

**Figure 1.** Non-dominated and dominated solutions.

A pretty wide range of developed algorithms may be found in the literature to solve the task graph scheduling problem, which indicates the great importance of the problem in engineering applications. The state-of-the-art algorithms in the literature mostly apply soft computing methods and metaheuristics for solving the problem [11–14]. For instance, SGA and EP methods were developed by authors in [14], as were MOGA and MOEP methods applied in [15]. Moreover, HEFT-NSGA, MFA, weighted sum MOEP, hybrid algorithms, and a multi-agent system were developed in [8,16–18], respectively. It can be seen that in the majority of state-of-the-art literature, a simple metaheuristic or weighted-sum method has been used. State-of-the-art methods are listed in Section 3. Likewise, Section 3 includes several new evolutionary algorithms proposed in recent literature.

This paper provides a novel hybrid method comprising the improved multi-objective differential evolution (MODE) method and variable neighborhood search (VNS) to schedule task graphs in distributed systems [19]. The novelty of the proposed hybrid method is achieved by improving MODE and hybridizing it with the VNS approach. The proposed method improves the performance of MODE by applying appropriate solution representation as well as effective selection, crossover, and mutation operators. Likewise, the VNS algorithm is applied to improve the extracted Pareto-front, speed up the evolution process, and increase the power of determining more promising parts of the search space. In the modified version of MODE, the selection operator is more effective due to applying roulette-wheel selection based on dominance rankings instead of fully random selections. The number of solutions dominating a solution is known as dominance rank, and consequently, the better solutions will have lower ranks. Therefore, the lower ranks are arranged to occupy larger parts of the roulette wheel to increase their selection chance (probability). Likewise, effective mutation and crossover operators are proposed in this paper to speed up the evolution process and increase the driven Pareto-front quality. A more promising portion of the search space is found in a novel proposed mutation because both task order and processors are modified without breaking the feasibility of the solution. Details of the innovative mutation and crossover operators are given in Section 4. In the modified MODE, non-dominated solutions found so far, Pareto-front, are kept in the archive and updated at the end of each MODE loop. Meanwhile, the proposed method applies a variable neighborhood search mechanism (VNS) over the archive after it is updated. This technique allows for more exploration and exploitation of the solutions in the archive to make them more accurate. However, to prevent time-consuming VNS, it is applied over a maximum of 10 solutions in the archive, and there are ten iterations of the inner loop. A description of the VNS method is given in Section 4.

Apart from the fact that DE is a straightforward optimization method, it is also robust and powerful. Like many other optimization methods, DE operates based on some parameters and several operators. The aim of optimization methods is to explore a high-quality PF in an acceptable time, preventing early convergence to avoid local optimal solutions. It is obvious that the quality of operators as well as the solution representation scheme affect the ability of DE to find better PF and speed up convergence. Therefore,

there is a rich literature to improve the performance of DE [20]. Likewise, hybridizing DE with other methods is another way to add more power to DE in discovering better PFs [21–23]. In our proposed method to optimize the well-known scheduling problem, not only operator improvement but also hybridization are applied to have a robust hybrid system to deal with objective functions (makespan, reliability, and flow time).

Evaluation of the proposed hybrid method's effectiveness and performance against well-known benchmarks gleaned from cutting-edge literature. In addition, the values of spacing and hyper-volume metrics are calculated. Furthermore, the Wilcoxon signed method is applied to carry out pairwise statistical tests over the obtained results. The proposed method exceeds all the state-of-the-art methods in terms of performance and quality of objective values, according to all findings and test results.

As future works, different optimization problems, e.g., task scheduling in cloud computing, digital twins, and the internet of things [24–26], can be solved by the proposed method. In addition, it is planned to replace the MODE algorithm with recently proposed optimization methods mentioned in Section 3 and see how they affect the performance of the system [27–29].

The remaining parts of the study are divided into five sections: The full definition of the multi-objective task graph scheduling problem is provided in Section 2. Likewise, Section 2 defines the objectives of the problem and related equations in detail. In Section 3, the most recent approaches to scheduling multi-objective task graphs are briefly discussed. In addition, some recent robust metaheuristics are reviewed in this section. Section 4 contains a description of the suggested unique hybrid technique in details. The section also represents the flowcharts and algorithms applied in the proposed system. The algorithm settings and experimental findings are reported and discussed in Section 5 to prove the high performance of the proposed hybrid system. Finally, Section 6 illustrates the study's findings as well as a few potential future research projects.

## 2. Multi-Objective Task Graph Scheduling Problem

In a multi-objective task graph scheduling problem, all the tasks of a directed acyclic task graph representing a parallel program are distributed over a fully connected heterogeneous distributed system in order to minimize the Makespan (total completion time), minimize the average flow time, and maximize the reliability. Instead of the reliability maximization, the reliability index is minimized in the literature. A task graph comprises some nodes representing the tasks and some directed edges indicating inter-processor communications [30]. The edges are weighted based on the communication cost between the processors when the tasks at the two ends of the edges are performed on different processors. The communication cost turns to zero if the tasks are executed on the same processor [30]. A sample task graph comprising 19 tasks is presented in Figure 2 including the name and time for each task [8,30]. The tasks are uniquely named by *t* followed by a number, and the number in the box on the right side of each task number is the task time.

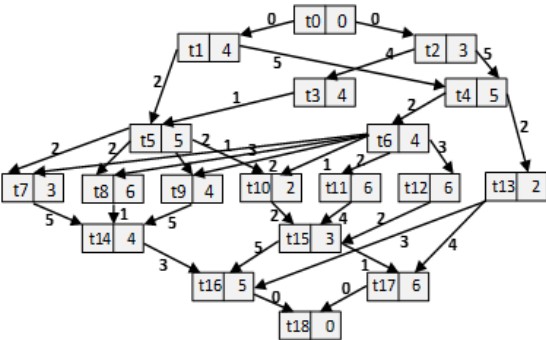

**Figure 2.** Sample Task Graph.

The goal of the task graph scheduling problem is to find an optimal schedule that maps tasks to the processors in a distributed system in order to optimize all the objectives.

Task graph scheduling problem is formulated as below [11–14]:

$$Min \ f = [f_1, f_2, f_3] \tag{1}$$

In Equation (1), $f_1, f_2$ and $f_3$ state the objective functions where $f_1$ indicates the total scheduling completion time that is known as Makespan. The value of $f_1$ is computed as below.

$$f_1 = max_j C_j(s) \tag{2}$$

$C_j(s)$ in Equation (2) is the time that processor $p_j$ finishes the execution and becomes idle. Consequently, $max_j C_j(s)$ indicates the completion time of the last processor in schedule $s$. The symbol 's' denotes the scheduling and it points to the scheduling represented by solution representation scheme in Figure 3.

| $t_0$ | $t_2$ | $t_3$ | $t_1$ | $t_4$ | $t_5$ | $t_6$ | $t_{11}$ | $t_{13}$ | $t_{12}$ | $t_9$ | $t_{10}$ | $t_8$ | $t_7$ | $t_{14}$ | $t_{15}$ | $t_{16}$ | $t_{17}$ | $t_{18}$ |
|---|---|---|---|---|---|---|---|---|---|---|---|---|---|---|---|---|---|---|
| $p_0$ | $p_1$ | $p_2$ | $p_0$ | $p_1$ | $p_2$ | $p_1$ | $p_1$ | $p_2$ | $p_1$ | $p_0$ | $p_0$ | $p_0$ | $p_2$ | $p_0$ | $p_1$ | $p_0$ | $p_1$ | $p_1$ |

**Figure 3.** A sample solution for graph given in Figure 2.

Total processor $p_j$ completion time is computed as Equation (3).

$$C_j(s) = \sum_{i \in v(j,s)} (st_{ij} + w_{ij}) \tag{3}$$

In Equation (3), all the tasks assigned to a processor $p_j$ belong to a set denoted by $v(j, s)$. Likewise, start and finish times of the task $v_i$ on processor $p_j$ are denoted by $st_{ij}$ and $w_{ij}$ respectively. In other words, $w_{ij}$ is the time processor $p_j$ finishes executing task $v_i$.

Second objective in Equation (1), $f_2$, indicates the average flow-time that is computed as Equation (4).

$$f_2 = aft(s) = \frac{\sum_j C_j(s)}{|P|} \tag{4}$$

The value of aft(s) in the schedule s is the summation of all completion times divided by $|P|$ (number of processors).

$f_3$ in Equation (1) denotes the value of the reliability index. It is important to notice that the reliability index minimization is equivalent to the reliability maximization [11,12]. There is a possibility for the processors to fail during the execution but failure of a processor does not affect the other processors. Probability of successfully performing all the tasks on processor $p_j$ is computed as Equation (5).

$$P_{succ}^j(s) = e^{-\lambda_j C_j(s)} \tag{5}$$

In Equation (5), $\lambda_j$ is the processor $p_j$'s failure rate. Eventually, Probability of successfully performing a schedule s is calculated as Equation (6).

$$P_{succ} = e^{-\sum_j \lambda_j C_j(s)} \tag{6}$$

Likewise, the communication reliability between the processors $p_m$ and $p_n$ is computed as Equation (7).

$$R_{mn}(V, S) = e^{-\lambda_{mn} \sum_{i=1}^{|V|} \sum_{j=1}^{|V|} s_{im} \cdot s_{jn} \cdot c_{ij}} \tag{7}$$

In Equation (7), the set of tasks is denoted by $V$ and the rate of communication failure rate between processors $p_m$ and $p_n$ is denoted by $\lambda_{mn}$. Meanwhile, $s_{im}$ and $s_{jn}$ indicate that tasks $i$ and $j$ have been mapped to processors $p_m$ and $p_n$. The value of $s_{im}$ is 1 if task $i$ has been scheduled to processor $m$. Likewise, $c_{ij}$ is the communication cost between task $i$ and $j$.

The Reliability index of schedule *s* is calculated as Equation (8). The number of the tasks is denoted by $|V|$.

$$f_3 = rel(s) = \sum_j \lambda_j C_j(s) + \sum_{m=1}^{|P|} \sum_{n=1}^{|P|} \sum_{i=1}^{|V|} \sum_{j=1}^{|V|} \lambda_{mn} s_{im} s_{jn} c_{ij} \qquad (8)$$

All the objectives $f_1$, $f_2$ and $f_3$ are conflicting; hence the optimal values of them cannot be achieved in a single solution [11–13]. Therefore, a set of non-dominated solutions, Pareto-front, is extracted for the multi-objective task graph scheduling problem.

## 3. Related Studies

In this part, the most recent approaches to the multi-objective task graph scheduling problem are briefly reviewed.

In [14], several algorithms were applied to a bi-objective (Makespan and reliability index) Gaussian elimination graph with 18 nodes by Chitra et al. They also evaluated the performance of the algorithms using the randomly generated graphs. the standard genetic algorithms (SGA) and evolutionary programming (EP) were applied by the authors, and they used a weighted-sum approach to combine the objectives into one objective. Moreover, the authors applied the multi-objective GA (MOGA) [30,31] and multi-objective evolutionary programming (MOEP) [32,33], and outcomes illustrated that the MOEP provides better distribution in Pareto-front than SGA, EP, and MOGA. Carrying out the comparison only between the GA and EP as well as using very small graphs can be taken into account as constraints in the study.

Chen et al. [17] suggested the HEFT-NSGA method to optimize the Makespan as well as the reliability index in the multi-objective task graph scheduling. Evaluations were carried out using some application graphs and random task graphs. the authors compared the outcomes with the Heterogeneous Earliest Finish Time (HEFT) method [34] and Critical Path GA (CPGA) [35] to illustrate that HEFT-NSGA extracts better solutions.

The multi-objective mean field annealing (MFA) is another metaheuristic used by Lotfi et al. [11] for solving the task graph scheduling problems. The authors evaluated their introduced method against the NSGAII [36] and MOGA [31] metaheuristics, and outcomes proved that the MFA extracts better Pareto-front in comparison to NSGAII and MOGA. The constraint of the study is that only very small graphs have been used for evaluation.

To solve the three-objective task graph scheduling problem, Chitra et al. [12] consumed the weighted sum GA [37], weighted sum MOEP [32], evolutionary programming (EP) [38,39], and MOGA [31] methods over a Gaussian elimination graph. There are no metrics calculations and no comparisons with state-of-the-art algorithms in the study.

For the bi-objective task graph scheduling problem, Eswari and Nickolas [40] proposed a firefly-based algorithm for optimizing Makespan and reliability solutions. In addition, comparisons and evaluations were done against modified GA (MGA) [41] and bi-objective GA (BGA) [42]. The findings indicated that the firefly method performs faster than MGA and BGA. Using a weighted-sum approach to merge objective values is the limitation of the suggested method. Likewise, no statistical analysis or metrics calculation can be found in [40].

Chitra et al. [14] merged multi-objective metaheuristics with a simple local search method to solve the bi-objective scheduling problem. SPEA2 and NSGAII metaheuristics were compared in their pure and hybrid versions.

Lotfi [8] proposed a strategy to combine six metaheuristics to solve two- and three-objective task graph scheduling problems. According to the proposed strategy, six metaheuristics collaborate and cooperate together to improve a shared population. The common population is divided into the subpopulations to be improved by metaheuristics, and all non-dominated solutions found so far are kept in a common archive. Also, each metaheuristic has its own local archive to keep non-dominated solutions during individual execution. Evaluations were done over different task graphs and compared to most of the state-of-the-art methods. Consequently, the evaluation results showed that the proposed ensemble method outperformed all considered competitors.

Likewise, there is a wide literature on new evolutionary algorithms that have been recently proposed by researchers [43–46].

Yanjiao et al. suggested the NSGA-II-WA algorithm to enhance the NSGAIII standard. When it comes to the evolution strategy and weight vector modification, their suggested NSGA-II-WA outperforms NSGAIII [43]. The suggested approach adds a discriminating condition, which speeds up the procedure without affecting performance. The effectiveness of the NSGA-II-WA in terms of convergence and distribution was tested by the authors using the DTLZ benchmark set [43].

In 2017, Xiang et al. introduced the VAEA (Vector Angle-Based Evolutionary Algorithm), which is based on angle decomposition [44]. Without the use of reference points, VAEA can adjust search space variety and convergence. The maximum vector-angle-first theory, used by VAEA, ensures that the solution set is wide and uniform. The findings of the authors' evaluation of VAEA using numerous objective benchmarks showed that VAEA effectively tunes convergence and diversity.

Cheng et al. suggested the RVEA (Reference Vector Guided Evolutionary Algorithm) in 2016 [46], which is based on reference vector guidance. RVEA tunes the weight vectors in accordance with objective functions dynamically. The authors compared the RVEA to five cutting-edge techniques and found that RVEA is efficient and effective.

Due to the wide range of state-of-the-art works in literature, it is useful to categorize the developed methods in terms of algorithm type and the problem type they are solving. The first categorization can be carried out in terms of the problem type, which is bi-objective or three-objective task graph scheduling problems. Likewise, the second organization is done based on the algorithm type, which can be either single-objective or multi-objective optimization approaches. The algorithms can also be either improved versions or hybrid types. Table 1 represents the categorization of state-of-the-art methods. Three recently proposed evolutionary algorithms are also considered in the table.

**Table 1.** Classification of the current methods in the literature.

| Method Title | Single-Objective Algorithm (Applying Weighted-Sum) | Multi-Objective Algorithm | Bi-Objective TGS | Three-Objective TSG | Reference Number |
|---|---|---|---|---|---|
| SGA | √ | ✗ | √ | ✗ | [15] |
| EP | √ | ✗ | √ | ✗ | [15] |
| Hybrid GA | √ | ✗ | √ | ✗ | [15] |
| EP | √ | ✗ | ✗ | √ | [12] |
| GA | √ | ✗ | ✗ | √ | [12] |
| MOGA | ✗ | √ | √ | √ | [14,31] |
| MOEP | ✗ | √ | √ | √ | [14,32] |
| HEFT | √ | ✗ | √ | √ | [17,34] |
| CPGA | √ | ✗ | √ | √ | [17,35] |
| HEFT-NSGA | ✗ | √ | √ | √ | [17] |
| MFA | ✗ | √ | ✗ | √ | [11] |
| FA | √ | ✗ | √ | ✗ | [13] |
| MGA | √ | ✗ | √ | ✗ | [42] |
| BGA | √ | ✗ | √ | ✗ | [41] |
| MOO+Local Search | ✗ | √ | √ | ✗ | [15] |
| Ensemble System | ✗ | √ | √ | √ | [8] |
| NSGA-II-WA | ✗ | √ | √ | √ | [43] |
| VAEA | ✗ | √ | √ | √ | [44] |
| RVEA | ✗ | √ | √ | √ | [46] |

The single-objective optimization algorithms use the weighted-sum method to be able to optimize more than one objective [47–49]. The reported results show that most of the multi-objective optimization methods outperform the single-objective and weighted-sum approaches.

## 4. The Proposed Hybrid Method

Efficiently solving the multi-objective multiprocessor scheduling problem is of great importance in engineering applications. The reason is that the task graph is the representation of a parallel program running over multiprocessors in distributed systems. Decreasing the total execution time and optimizing other objectives of the problem play a remarkable role in distributed systems, and this is the reason why the problem has been widely solved by many different approaches up to now and is still going on. The new hybrid method that combines the improved MODE algorithm [19,50,51] and VNS methodology [52,53] is described in this part. According to [51], DE is a simple, effective, and robust algorithm for solving global optimization problems. Many research efforts have been made to improve DE and apply it to different practical problems. Differential evolution is able to search a very large space of candidate solutions, and its biggest advantage is stability. Since the DE is simple, robust, and stable, it was selected as the main method to be hybridized with another fast and efficient search technique called VNS. Applying the pure MODE algorithm is not promising; the selection, crossover, and mutation operators in MODE are therefore modified and improved in this paper. The dominance rank to be used in the selection part affects the performance in a good way. Likewise, to increase the performance, MODE has been hybridized with a fast and robust VNS. These are the motivations for this paper to merge MODE and VNS in order to reach a reliable and robust method. With regard to selection, crossover, and mutation operators, the suggested hybrid technique makes use of a modified MODE. All non-dominated solutions found so far, Pareto-front, are kept in an archive, which is updated at the end of each cycle in MODE.

The population is randomly initialized at the beginning of the proposed method. A solution (scheduling) is represented in the suggested way as an array including two rows and n columns, in which n is the total number of tasks in the graph. As an example, Figure 3 provides a random schedule for the graph seen in Figure 2. The $t_i$ symbol is used to point the task number assigned to processor $p_j$ explained in Section 3. Hence, each column in array indicates the assignment of task $t_i$ to processor $p_j$ e.g., the second column illustrates that $t_2$ has been assigned to $p_1$.

The procedure depicted in Algorithm 1 is used to randomly initialize the population. While the tasks are determined based on the topological ordering, the processors are chosen at random. The ability of this algorithm to provide diverse, legitimate, random solutions—solutions that differ in terms of task ordering and processors—is one of its advantages. The algorithm retains a list of all ready tasks to be done, called ReadyTasks, and it selects tasks at random one by one from this list to generate solutions. If all of a task's parents have previously been completed, the task is ready to be executed. For this, the algorithm gives each job a parent counter and decrements it whenever a parent is executed. In this manner, when the parent counter reaches zero, a task is added to the ReadyTasks list. The Successors of a task $t_i$ in the algorithm is the set of its children on task graph e.g., the successors of $t_5$ on graph shown in Figure 2 are {$t_7$, $t_8$, $t_9$, $t_{10}$}.

The proposed hybrid method for tackling the mentioned problems is shown in Figure 4. As can be seen in the flowchart, the system continues to operate in successive sessions until the termination requirements are satisfied. The system first determines the values of each objective using the formulae and explanations from Section 2. The algorithm for calculating a schedule's makespan is shown in Algorithm 2. The AT[$t_i$] and FT[$t_i$] variables in the algorithm represent the readiness of each task to begin execution and its completion, respectively. Also, the $P[p_i]$ records when the processor $p_i$ will be free. Moreover, $|P|$ and $|T|$ stand for the number of processors and tasks, respectively. The value of communication_time ($t_i$, $t_j$) also indicates the cost of communication between

the processor executing $t_i$ and the processor executing $t_j$. It should be noted that if both processors are same, the communication cost will be zero. Finally, the maximum $P[p_i]$ for all processors is considered to determine the makespan of the schedule.

---

**Algorithm 1:** Schedule Initialization Algorithm

---

Schedule-Initialization (schedule [1 . . . 2][1 . . . n], *V*, *P*) // *V is the set of tasks, //P is the set of processors*

　　For all Tasks $t_i \in V$ in task graph
　　　　　ParentsCount [$t_i$] = number of $t_i$ parents in task graph
　　ReadyTasks = {$t_i \in V$ | ParentsCount [$t_i$] = 0} // *Prepare the ready tasks to execute*
　　$j = 1$
　　While (ReadyTasks set is not Empty)
　　　　　Choose a Task $t_k$ from the ReadyTasks set randomly
　　　　　Add $t_k$ to Schedule [1][*j*]
　　　　　Choose a Processor p from the ProcessorList randomly
　　　　　Add p to Schedule [2][*j*]
　　　　　$j = j + 1$
　　　　　For all Children $t_i \in$ {Successors of $t_k$}
　　　　　　　ParentsCount [$t_i$] = ParentsCount [$t_i$] $-$ 1
　　　　　　　if (ParentsCount [$t_i$] == 0)
　　　　　　　　　Add $t_i$ to ReadyTasks set

---

The dominance rank of each solution is determined in the following phase, where s is the number of solutions that are dominating it. The lower rank value indicates the better quality of the answer. Dominance rank values are used in roulette wheel selection, in which good solutions are more likely to be selected than poor solutions. The roulette wheel selection mechanism selects the solutions according to the size of the region they occupy on the wheel. Hence, the value of dominance ranks is changed in such a way that big values show better ranks. To do this, all dominance ranks are subtracted from the biggest dominance rank.

Later on, the archive is updated with recently found non-dominated solutions, according to the flowchart in Figure 4. This phase also uses dominance rank values, in which it inserts all solutions with dominance rank zero into the archive and then removes all solutions in the archive dominated by the newly inserted one.

---

**Algorithm 2:** Makespan Calculation

---

Makespan-Calculation (Schedule [1 . . . 2][1 . . . n], ExecutionTime [], CommunicationTime [])
　　　　　　　　　　　　　　*//ExecutionTime is the tasks execution time*
　　　　　　　　　　　　　　*//CommunicationTime is the cost of edges between task pairs*
$P$ [1|*P*|] = {0}, AT [1 . . . |T|] = {0}, FT [1 . . . |T|] = {0}
　　　*// |P| and |T| are the number of processors and number of tasks respectively*
　　　*//P[pi] is the time at which processor pi becomes idle*
　　　*// AT[$t_i$]is the time that ti would be ready to execute*
　　　*// FT[$t_i$]is the finish time of task$t_i$*
for $i = 0$ to |T|
　　　$t_i$ = Schedule [0][*i*]
　　　$P$ [Schedule [1][*i*]] = max (AT [$t_i$], $P$ [Schedule [1][*i*]] + ExecutionTime($t_i$))
　　　FT[$t_i$] = $P$ [schedule [1][*i*]]
　　　for all Tasks $t_j \in$ Successors($t_i$) in the task graph
　　　　　temp = FT [$t_i$];
　　　　　if (schedule [1][*i*] is not same as processor assigned to $t_j$)
　　　　　　　temp = temp + Communication_time ($t_i$, $t_j$)
　　　　　AT [$t_j$] = max (temp, AT [$t_j$])
　　　Makespan = Max ($P$ [1 . . . |*P*|])

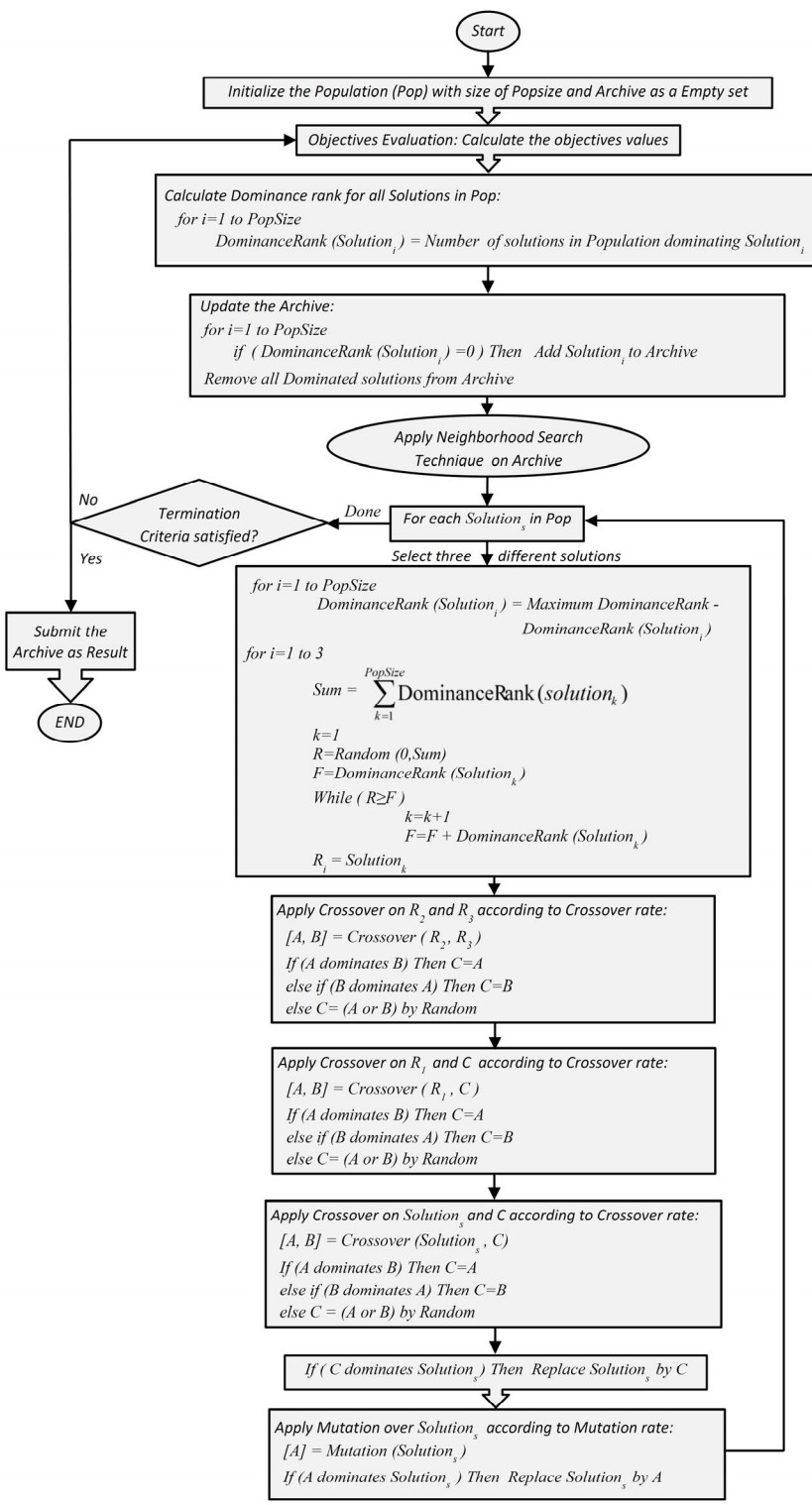

**Figure 4.** Proposed Hybrid Method.

In the next step, the VNS method is used across archives to further leverage the greatest solutions so far discovered. In this manner, the non-dominated solutions in the archive are modified and improved. Algorithm 3 shows the pseudocode for the VNS algorithm. The suggested hybrid method applies the VNS methodology over a maximum of 10 solutions in the archive and iterates the inner loop 10 times to prevent time-consuming VNS. The definition of the neighborhood structure N in the VNS algorithm results in a moderately significant alteration of the solution.

---

**Algorithm 3:** VNS method

---

VNS (Archive) // *Archive consists of all non-dominated solutions found so far*
Define a neighborhood structure // *It is a modification way to change a solution*
// *The modification is performed using the mutation operator presented in Figure 9*
While (VNS has not been applied on 10 solutions)
      Choose a random solution X from archive
      for k = 1 to 10
            Generate a solution Y from X using the structure N
            for p = 1 to 3 // *Local Search is applied on solution Y*
                  Generate a new solution Z from Y by changing 3 processors randomly
                if (Z dominated Y)
                    Copy Z to Y
              if (Y dominates X)
                Copy Y to X

---

The hybrid algorithm then has an inner loop that uses the DE to perform the exploration task. For each solution among the population, the loop iterates the following steps again. Three potential solutions are chosen in the first stage using a roulette wheel selection method while considering the dominance rank values. Since the roulette wheel selection mechanism gives more chances to the bigger values, it will more likely select the solutions with a higher dominance rank. To prevent this, all dominance rank values are subtracted by the biggest dominance rank in the population. This way, the rank of the worst solution becomes zero, and for the other solutions, the higher dominance rank indicates a better solution. Therefore, the selection step chooses three random solutions for each solution in the population so that better solutions have a better chance of being selected. In Figure 4, the algorithm is presented. To identify the final solution C, the crossover operator is then used over three solutions that were arbitrarily chosen in the following three steps. Then solution *i* is replaced by solution C if the solution C dominates solution *i*. Crossover operators are implemented in a way that results in workable solutions. Only processors are used for this two-point crossover. In this manner, only the processors are joined, and the order of the jobs is maintained because tasks follow a topological order and any random combination would break the feasibility of the solution. Meanwhile, the crossover operator is applied according to the crossover rate, which has been defined between 0 and 1. Algorithm 4 indicates the crossover algorithm. In the algorithm, random (0, 1) generates a uniformly distributed random number in the interval (0, 1). Likewise, Cutpoint1 and Cutpoit2 should be generated under the condition that Cutpoint1 must be smaller than Cutpoit2.

---

**Algorithm 4:** Crossover

---

Crossover (Parent1 [1 . . . 2][1 . . . n], Parent2 [1 . . . 2][1 . . . n])
R = random (0, 1) // *Generate a random number between 0 and 1 for Crossover Rate*
If (R < CrossoverProbability)
      Cutpoint1 = RandomNumber (1, n)
      Cutpoint2 = RandomNumber (1, n)
            For *i* = 1 to Cutpoint1
      Swap (Parent1 [2][*i*] and Parent2 [2][*i*])
            For *i* = Cutpoint2 to n
      Swap (Parent1 [2][*i*] and Parent2 [2][*i*])

---

The mutation operator is applied to the solution I in the inner loop's final step. The sequence of the jobs is also changed by the suggested approach for the mutation operator. The algorithm selects a point on the solution between 1 and the number of tasks randomly. The order of the jobs is then kept unchanged up until that random point, but after that point, the order is changed arbitrarily. The adjustment is implemented using the procedure shown in Algorithm 1, in which the remaining portion of the solution is randomly generated after the given point. The benefit of the suggested mutation is that the solution created by this

operator is altered in terms of both tasks and processors, which causes the algorithm to hop through the search space and find better solutions. The algorithm also picks a few CPUs and switches them at random. In Algorithm 5, the mutation algorithm is displayed.

---

**Algorithm 5:** Mutation

---

Mutation (Schedule [1 . . . 2][1 . . . n], $V$) // $V$ is the set of tasks
NewSchedule = Schedule // NewSchedule is mutated version of Schedule
For all Tasks $t_i \in V$ in task graph //Count the number of parents for each task
      ParentsCount [$t_i$] = number of $t_i$ parents in task graph
ReadyTasks = {$t_i \epsilon V$ | ParentsCount [$t_i$] = 0} // *Prepare the ready tasks to execute*
*ReadyCount = Number of tasks in ReadyTasks set*
*p = 0, pp = 0, cutpoint = RandomNumber (1, n)*
*q = Random (1, cutpoint) // After cutpoint, the order of tasks will be changed randomly*
*While (ReadyCount >= 0)*
      SelectCount = *Number of tasks in ReadyTasks set*
      SelectList = ReadyTasks
      If (SelectCount > 1)
            pp = pp + 1
      If (pp >= q) // if it is after cutpoint, the next task is selected randomly amongst ready tasks
            s = Random (1, SelectCount)
            t = SelectList (s) //choose a task from ready tasks randomly
            Remove t from ReadyTasks
            ReadyCount = ReadyCount − 1
            p = p + 1
            NewSchedule [1][p] = t
      Else // if it is before cutpoint, the next task is selected from Schedule
            p = p + 1
            t = Schedule [1][p]
            ReadyCount = ReadyCount − 1
      For all Children $c_i \in$ {Successors of t}
            ParentsCount [$c_i$] = ParentsCount [$c_i$] − 1 //decrement the number of parents by
one
                  If (ParentsCount [$c_i$] == 0)
                        Add $c_i$ to ReadyTasks set //add new ready tasks to ReadyTasks set
For i = 1 to 3 //exchange the processors three times
      R1 = Random (1, n);
      R2 = Random (1, n);
SWAP (NewSchedule [2] [R1] and solution [2] [R2]);

---

If the termination requirements are not met when the inner loop is terminated, the hybrid method moves on to the next session. Otherwise, the extracted archive is submitted as the best Pareto front found so far.

## 5. Results and Discussion

The evaluation of the developed algorithm is carried out in this section by taking well-known benchmarks from the related literature. Whereas applying the pure MODE algorithm is not promising, this paper modifies it in terms of selection and crossover operators. The dominance rank used in the selection part significantly affects performance. Likewise, to increase performance, MODE is hybridized with a fast and robust neighborhood search technique. Consequently, the results become promising enough to extract high-quality solutions.

### 5.1. Parameter Values

The values of all parameters regarding the MODE algorithm are given in Table 2. In our Matlab® implementation, the generation size and processor number are adjusted according to the literature. Population size and generation size are two effective parameters

influencing time complexity. The values of these parameters are set similarly to the state-of-the-art methods to make the comparison fair. As it can be seen in Table 2, the population size and generation size are set to 200 and 300, respectively. It is worth mentioning that the VNS technique does not add much additional time complexity to the hybrid method because the small and fast version of VNS is applied. As a result, the suggested hybrid strategy is more effective considering the quality of the Pareto front.

**Table 2.** Parameters values.

| Algorithm | Parameter Values | | | | |
|---|---|---|---|---|---|
| MODE | \|Pop\| | Scaling_Factor | #of Generations | PC | PM |
| | 200 | 0.5 | 300 | 0.8 | 0.4 |

### 5.2. Performance Evaluation Using Bi-Objective Benchmarks

To evaluate the performance, two well-known metrics are calculated. Since the optimal PF (Pareto-Front) is unknown for the benchmarks, it is not possible to compute all the metrics. The spacing metric is computed as follows to evaluate the diversity of the PF [14]. Likewise, the hypervolume metric evaluates the convergence and spread of PF [31]. Meanwhile, in the evaluations, one of the recently proposed algorithms, NSGA-II-WA, and the ensemble method proposed in [8] are taken into account as competitors. The Gaussian Elimination Graph (GE) is the first benchmark, which is shown in Figure 5.

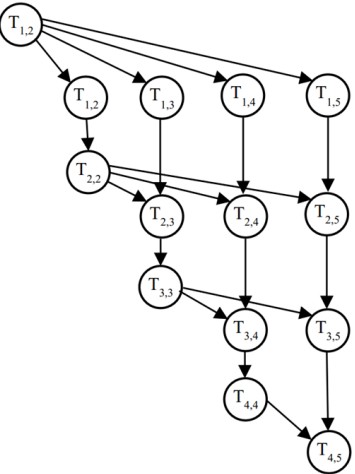

**Figure 5.** Gaussian Elimination Graph for size = 5.

A graph with 10 and 54 nodes was taken into consideration in [13]. Also, maintenance and reliability were considered objectives.

The results obtained by the proposed hybrid method and five competitors (Bi-objective GA (BGA) [42], Modified GA (MGA) [41], Firefly-based algorithm (FA) [13], NSGA-II-WA, and Ensemble System [8]) are represented in Table 3. The results are calculated according to the different values of CCR (1, 5, and 10). The following equation indicates the way to compute the value of CCR:

$$CCR = \frac{Average\ Communication\ Cost}{Average\ Computation\ Cost} \tag{9}$$

It should be pointed out that BGA, MGA, and FA methods adapt the multi-objective problem to a single-objective problem using a weighted-sum approach, but NSGA-WA, the ensemble method, and the proposed hybrid method extract the Pareto front. Figures 6–8 show the Pareto front extracted by the proposed hybrid method for different CCR values.

**Table 3.** Obtained Objective Values.

| Objective | Method | CCR = 1 | CCR = 2 | CCR = 3 |
|---|---|---|---|---|
| Makespan | FA | 458 | 687 | 1144 |
| | MGA | 591 | 1070 | 1426 |
| | BGA | 616 | 1103 | 1490 |
| | NSGA-II-WA | 433 | 841 | 1065 |
| | Ensemble System | 420 | 657 | 1069 |
| | Hybrid Method | 418 | 642 | 1055 |
| Reliability index | FA | 9.45 | 7.4 | 15.48 |
| | MGA | 13.17 | 15.93 | 23.54 |
| | BGA | 9.48 | 12.20 | 22.47 |
| | NSGA-II-WA | 10.56 | 8.30 | 16.66 |
| | Ensemble System | 8.29 | 6.76 | 14.83 |
| | Hybrid Method | 8.05 | 6.61 | 13.40 |

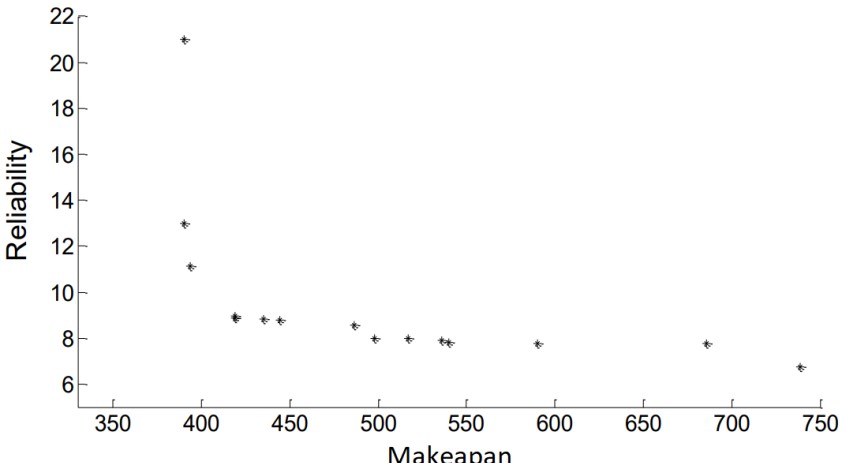

**Figure 6.** Extracted PF for CCR = 1.

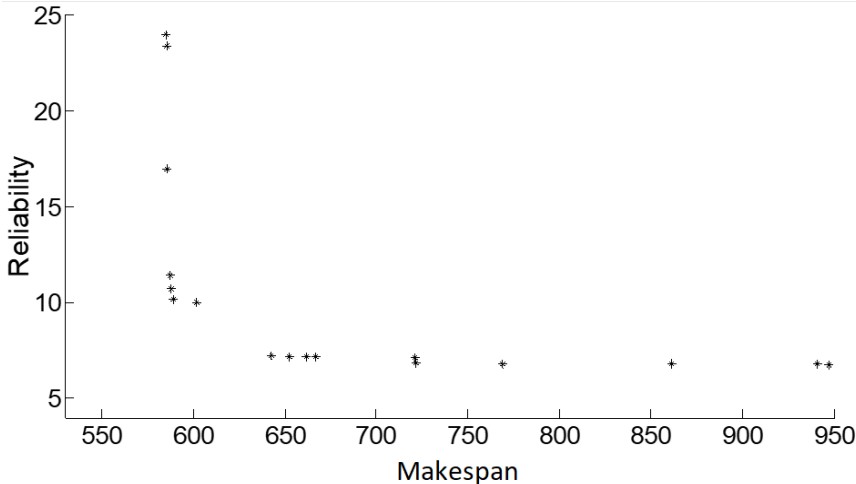

**Figure 7.** Extracted PF for CCR = 2.

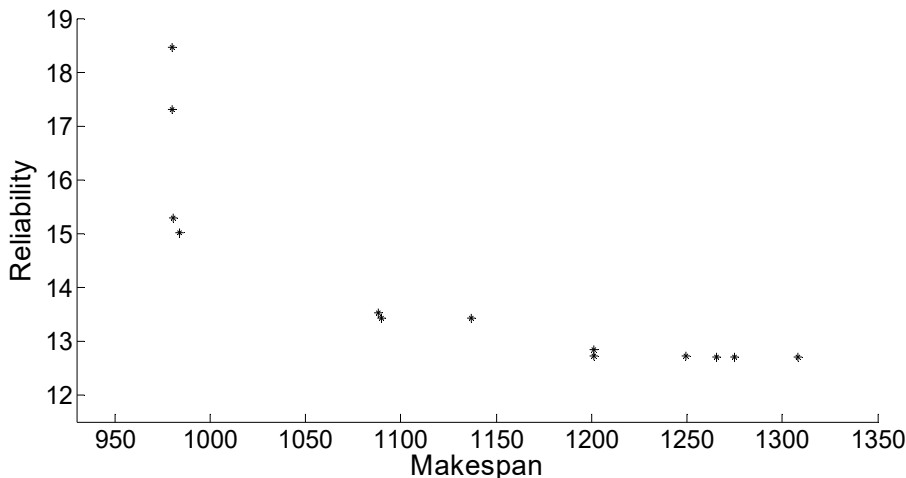

**Figure 8.** Extracted PF for CCR = 3.

Table 3 indicates that the proposed method produces better results than its competitors in terms of reliability and time span. The spacing and hypervolume values are represented in Table 4.

**Table 4.** Spacing and Hypervolume of obtained PF.

| CCR | Spacing | Hypervolume |
|---|---|---|
| 1 | 35.65 | 0.91823 |
| 2 | 28.90 | 0.883519 |
| 3 | 31.26 | 0.940621 |

### 5.3. Performance Evaluation Using Three-Objective Benchmarks

The nonparametric Wilcoxon signed rank test is carried out by following the procedure in [44] to confirm that the results are different. In Table 5, the sum of all better ranks and the sum of all worse ranks are represented by R+ and R−, respectively. In order to confirm the rejection of the null hypothesis, the significance level ($\alpha$) and the *p*-values are calculated. Consequently, due to the *p*-value being less than 1, the null hypothesis is rejected.

**Table 5.** Wilcoxon signed test results.

| Method | R+ | R− | $\alpha$ | *p* Value |
|---|---|---|---|---|
| FA | 78 | 25 | 0.05 | 0.002364 |
| MGA | 62 | 18 | 0.01 | 0.000231 |
| BGA | 81 | 12 | 0.01 | 0.000843 |
| NSGA-II-WA | 71 | 25 | 0.01 | 0.091024 |
| Ensemble System | 61 | 28 | 0.05 | 0.115243 |
| Hybrid Method | 56 | 32 | 0.06 | 0.325524 |

The second experiment is done over the benchmarks reported in [15]. The proposed hybrid method is compared to evolutionary programming, hybrid GA (HGA), GA, NSGA-II-WA, and ensemble systems. Table 6 illustrates the obtained results and shows that the proposed method outperforms its competitors.

**Table 6.** Obtained objective values.

| Objective | Method | Makespan | Reliability Index |
|---|---|---|---|
| Best Makespan | GA | 584 | 14.88 |
| | EP | 594 | 15.77 |
| | HGA | 562 | 13.37 |
| | NSGA-II-WA | 511 | 11.23 |
| | Ensemble System | **471.23** | **8.85** |
| | Hybrid Method | 468.92 | 8.73 |
| Best Reliability index | GA | 961 | 6.64 |
| | EP | 964 | 7.19 |
| | HGA | 1243 | 4.35 |
| | NSGA-II-WA | 680.37 | 4.03 |
| | Ensemble System | 661.43 | 3.62 |
| | Hybrid Method | **648.25** | **3.62** |

In addition, Table 7 confirms the rejection of the null hypothesis and the significant differences between the results.

**Table 7.** Wilcoxon signed test results.

| Method | R+ | R− | $\alpha$ | *p*-Value |
|---|---|---|---|---|
| GA | 62 | 19 | 0.035 | 0.002938 |
| EP | 49 | 28 | 0.042 | 0.007328 |
| HGA | 32 | 21 | 0.032 | 0.006401 |
| NSGA-II-WA | 41 | 32 | 0.041 | 0.008324 |
| Ensemble System | 44 | 29 | 0.052 | 0.019232 |
| Hybrid Method | 36 | 27 | 0.057 | 0.029351 |

The next evaluation is carried out based on the benchmarks and results reported in [12]. The proposed hybrid method is compared to EP, GA, NSGA-II-WA, and a system to solve a three-objective multi-processor scheduling problem, where the objectives are makespan, mean flow time, and reliability. Table 8 represents the results gained by three methods.

Table 8 indicates that the proposed hybrid method performs better than GA and EP. Also, Figure 9 represents the extracted Pareto-Front consisting of 73 solutions by the proposed hybrid method. Table 8's values were chosen at random from the Pareto front. The derived Pareto-front has spacing and hypervolume values of 18.32 and 0.78921, respectively.

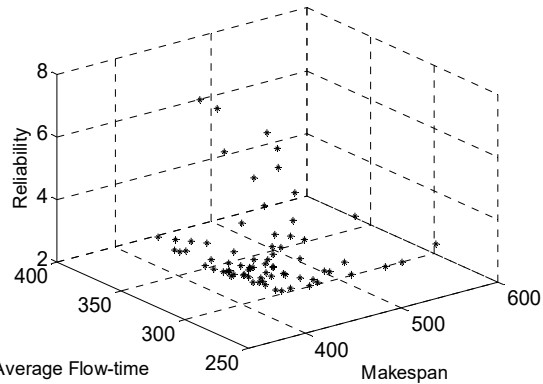

**Figure 9.** Pareto Front found by developed hybrid method.

**Table 8.** Obtained objective values.

| Objective | Method | Makespan | Reliability Index | Flow Time |
|---|---|---|---|---|
| Best Makespan | GA | 416 | 372.75 | 7.18 |
| | EP | 419 | 368.5 | 7.02 |
| | NSGA-II-WA | 412 | 325.43 | 6.85 |
| | Ensemble System | 404.10 | 303.59 | 5.18 |
| | Hybrid Method | **398.45** | **301.34** | **4.41** |
| Best Reliability index | GA | 603 | 280.75 | 5.6 |
| | EP | 632 | 292 | 5.6 |
| | NSGA-II-WA | 511.60 | 308.16 | 4.23 |
| | Ensemble System | 483.88 | 266.06 | 3.81 |
| | Hybrid Method | **461.02** | **242.16** | **3.11** |
| Best Average FlowTime | GA | 810 | 281.25 | 3.07 |
| | EP | 818 | 284.55 | 3.10 |
| | NSGA-II-WA | 659.63 | 308.41 | 3.49 |
| | Ensemble System | 502.76 | 276.34 | 2.83 |
| | Hybrid Method | **502.76** | **258.55** | **2.62** |

The authors of [11] compared their suggested MFA approach to NSGAII using the test graph shown in Figure 2 on four CPUs. They evaluated the problem objectives to be makespan, mean flow time, and reliability. Table 9 illustrates the results obtained by the developed hybrid methods: MFA, NSGAII, NSGA-II-WA, and Ensemble System.

**Table 9.** Obtained objective values.

| Objective | Method | Makespan | Reliability Index | Flow Time |
|---|---|---|---|---|
| Best Makespan | NSGAII | 62 | 25 | 53 |
| | MFA | 59 | 25 | 51 |
| | NSGA-II-WA | 54 | 25 | 51 |
| | Ensemble System | 52 | 21 | 53 |
| | Hybrid Method | **50** | **22** | **51** |
| Best Reliability index | NSGAII | 65 | 24 | 49 |
| | MFA | 61 | 24 | 50 |
| | NSGA-II-WA | 61 | 24 | 49 |
| | Ensemble System | 59 | 19 | 46 |
| | Hybrid Method | **60** | **18** | **47** |
| Best Average FlowTime | NSGAII | 65 | 24 | 49 |
| | MFA | 61 | 24 | 50 |
| | NSGA-II-WA | 59 | 24 | 47 |
| | Ensemble System | 59 | 21 | 43 |
| | Hybrid Method | **58** | **21** | **41** |

It can be seen that the proposed method outperforms MFA and NSGAII. The Pareto front extracted by the proposed hybrid method is represented in Figure 10. In addition, the derived Pareto-front has spacing and hypervolume values of 42.12 and 0.96547, respectively.

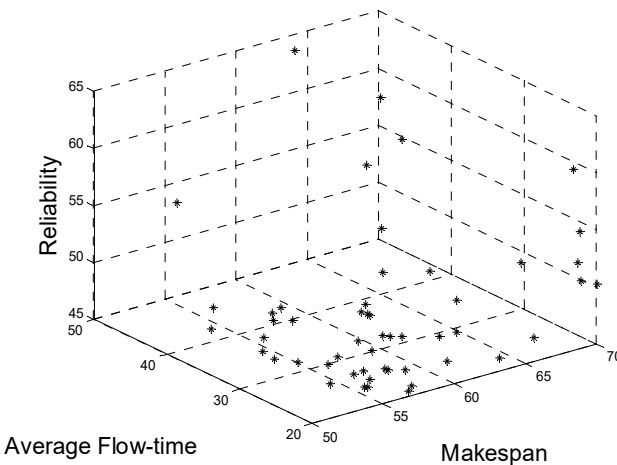

**Figure 10.** Pareto Front extracted by proposed hybrid method.

The results of the final assessment are provided in [17]. The HEFT [34] and CPGA algorithms [35] were contrasted with the recommended technique, HEFT-NSGA [17]. Findings are presented on the Fast Fourier Transformation and Gaussian Elimination graphs [54–56]. The FFT graph is shown in Figure 11. Schedule Length Ratio (SLR), another comparative parameter taken into account in [19], is computed as follows:

$$SLR = \frac{Makespan}{\sum_{v_j \in CP_{min}} \min(cost(v_j))} \tag{10}$$

where *CP* stands for the graph's critical path, which is focused on minimizing computing costs. The minimal computing costs of tasks on the critical path are added up to form *SLR*. The best algorithm is one that has the lowest *SLR*. The reliability values for the proposed hybrid technique over the Gaussian Elimination graph on four processors are shown in Figure 12 for the HEFT, CPGA, HEFT-NSGA, and HEFT-NSGA algorithms.

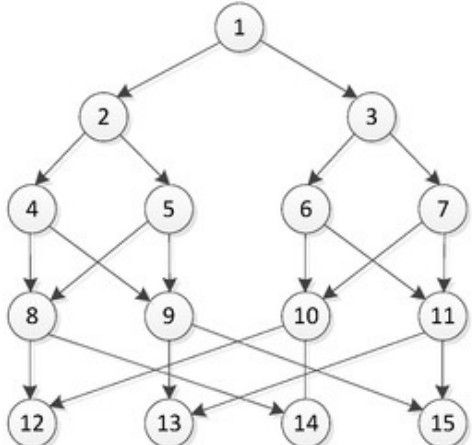

**Figure 11.** FFT Graph.

Figure 13 shows the Makespan, SLR, and reliability values for HEFT, CPGS, and HEFT-NSGA methods and depicts hybrid methods over the FFT graph.

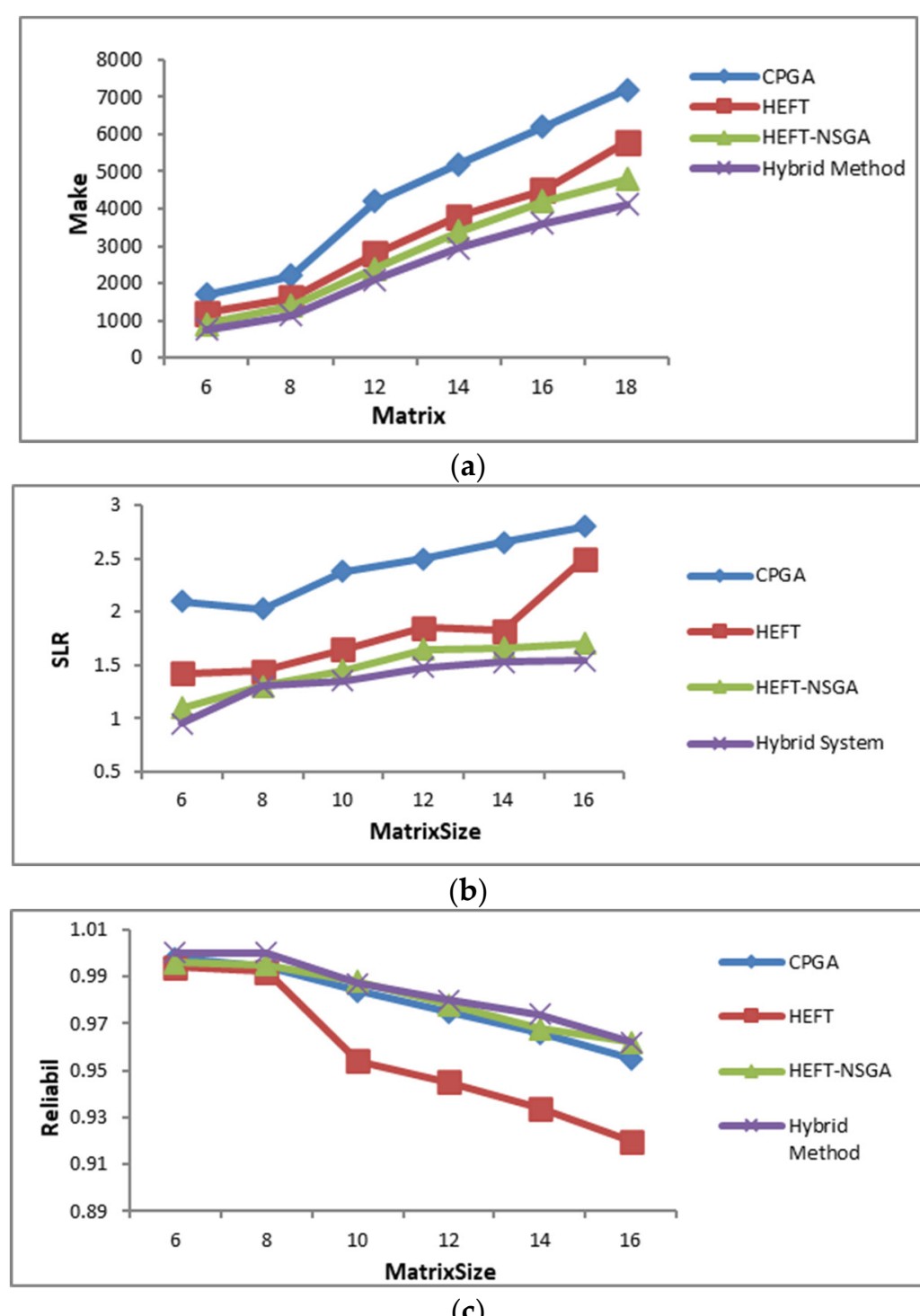

(**a**)

(**b**)

(**c**)

**Figure 12.** Makespan (**a**), SLR (**b**) and Reliability index (**c**) of CPGA, HEFT, HEFT-NSGA and proposed Hybrid method for Gaussian Elimination Graph.

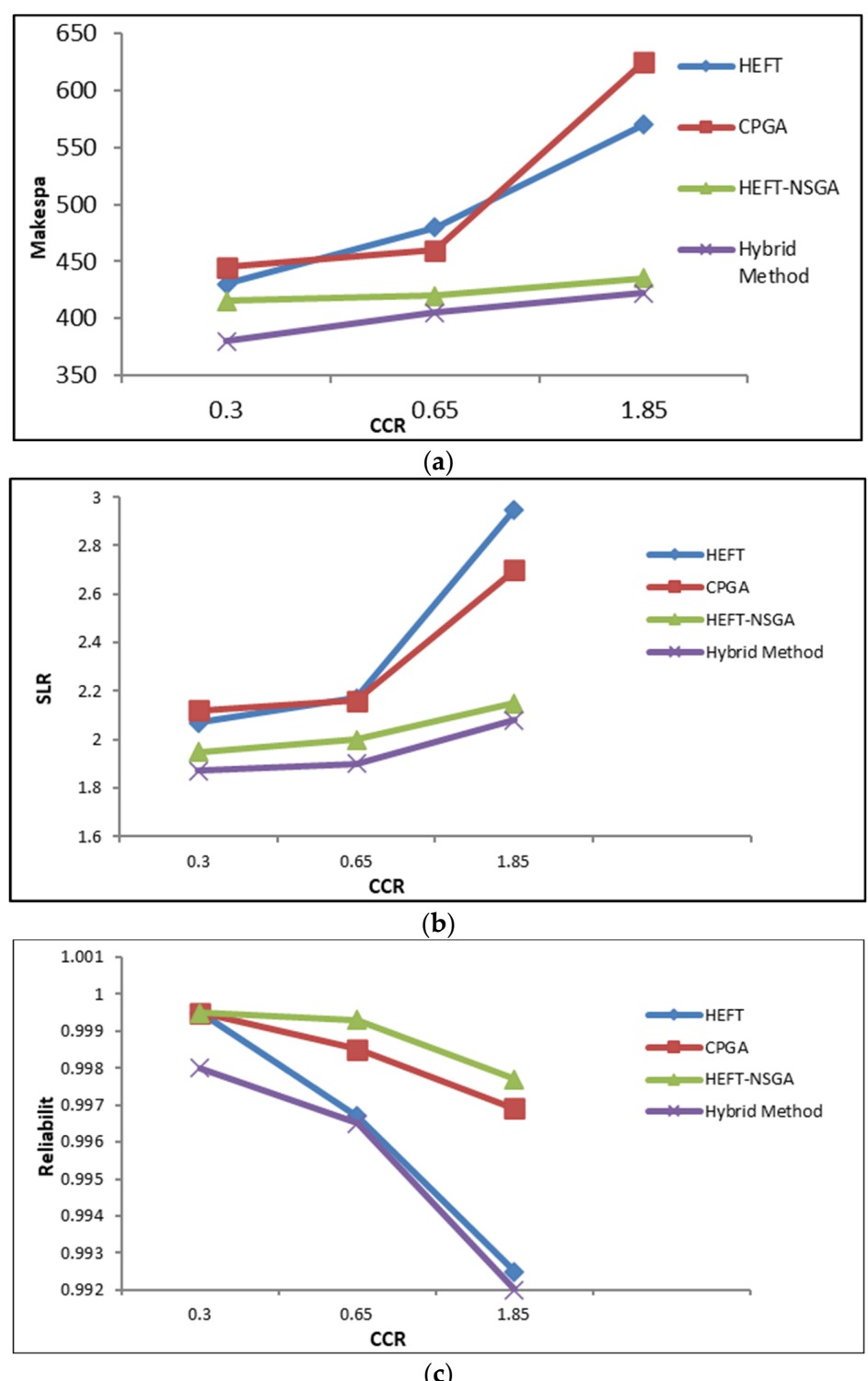

**Figure 13.** Makespan (**a**), SLR (**b**) and Reliability index (**c**) of CPGS, HEFT, HEFT-NSGA and proposed Hybrid method for FFT Graph.

## 6. Conclusions

This article offers a novel hybrid approach to the multi-objective problems of multi-processor scheduling in heterogeneous contexts. The proposed method relies on a strategy to combine a modified version of the MODE method with a variable neighborhood search technique. The novelty of this study is to modify the differential evolution method and combine it with neighborhood search to increase the ability to notice more promising portions

of the search space. To create a novel method, MODE's effective selection, crossover, and mutation operators are changed. Instead of using random selections, the selection operator is based on dominance rank values to boost the likelihood of selecting better solutions. As a result, superior solutions will be chosen more frequently than subpar ones in terms of dominance. However, to maintain diversity, poor solutions have a lower chance of being selected in this method. Crossover and mutation operators are performed according to proposed methods to increase the power of exploration and exploitation. The proposed novel mutation operator is done in such a way that it is able to change both task order and processors without breaking the feasibility of the solution. In addition, a quick variation of the variable neighborhood search strategy is used in the algorithm to get more precise results in the archive. It can be clearly seen from Table 9 that the proposed system has obtained 50, 18, and 41 for span, reliability, and flow time, respectively, which are better than the other competitors in the table.

Further studies may be conducted with the use of the developed algorithm in different scenarios for optimization problems, e.g., task scheduling in manufacturing and/or cloud computing [57–62], as discussed in Section 3. As well, it is planned to replace the MODE algorithm with recently proposed optimization methods, e.g., [43,44], and see how it affects the performance of the system.

**Author Contributions:** Conceptualization, N.L.; methodology, N.L.; software, N.L.; validation, M.G.N.; formal analysis, M.G.N.; investigation, N.L. and M.G.N.; resources, M.G.N.; data curation, N.L.; writing—original draft preparation, N.L. and M.G.N.; writing—review and editing, N.L. and M.G.N.; visualization, N.L. and M.G.N.; supervision, N.L.; project administration, N.L. and M.G.N. All authors have read and agreed to the published version of the manuscript.

**Funding:** This research received no external funding.

**Institutional Review Board Statement:** Not applicable.

**Informed Consent Statement:** Not applicable.

**Data Availability Statement:** Data is available according to the request.

**Conflicts of Interest:** The authors declare no conflict of interest.

## Nomenclature List

| | |
|---|---|
| AT[$t_i$] | Readiness of each task to begin execution |
| aft(s) | The summation of all completion times divided by \|$P$\| |
| BGA | Bi-objective Genetic Algorithm |
| CPGA | Critical Path Genetic Algorithm |
| Cj(s) | The time that processor p$_j$ finishes execution |
| DE | Differential Evolution |
| EP | Evolutionary Programming |
| FA | Firefly based Algorithm |
| FT[$t_i$] | The Completion time of task $i$ |
| GA | Genetic Algorithm |
| GE | Gaussian Elimination Graph |
| HEFT | Heterogeneous Earliest Finish Time |
| $Max_j C_j(s)$ | Completion time of last processor in schedule $s$ |
| MODE | Multi-objective Differential Evolution |
| MOEP | Multi-objective Evolutionary Programming |
| MFA | Mean Field Annealing |
| MOGA | Multi-objective Genetic Algorithm |
| MOO | Multi-objective Optimization |
| NP | Non-deterministic Polynomial |
| NSGAII | Non-Dominated Sorting Genetic Algorithm |
| PF | Pareto-Front |
| RVEA | Reference Vector guided Evolutionary Algorithm |
| R+ | Sum of all better ranks |

| | |
|---|---|
| R− | Sum of all worse ranks |
| SGA | Standard Genetic Algorithm |
| $S_{im}$ | mapping task $i$ to processor $p_m$ |
| $S_{jn}$ | mapping task $j$ to processor $p_n$ |
| VNS | Variable Neighborhood Search |
| VAEA | Vector Angle-Based Evolutionary Algorithm |
| $v(j, s)$ | All tasks assigned to processor $p_j$ |
| #p | The number of tasks |
| #t | The number of processors |
| $\alpha$ | Significance level |
| $\lambda_j$ | The failure rate of processor $p_j$ |
| $\lambda_{mn}$ | The communication failure rate of processors $p_m$ and $p_n$ |

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
