# Peer review of "A New Hybrid Algorithm Based on Improved MODE and PF Neighborhood Search for Scheduling Task Graphs in Heterogeneous Distributed Systems"

_applsci, doi:10.3390/app13148537_

Round 1
Reviewer 1 Report
The proposed hybrid method is effective and performs against well-known benchmarks gleaned from cutting-edge literature. The author calculated the values of spacing and hyper-volume metrics. The Wilcoxon signed method is applied to carry out a pairwise statistical test over obtained results. The proposed method exceeds all state-of-the-art methods in terms of performance and quality of objective values, according to all findings and test results.
The paper is attractive and has enough novelty for publication
Recommendation: Minor
1. Remove the typo mistakes
2. The author should add one more section of Nomenclature before the references section.
3. The reference list should be according to the journal format
4. Align the equations and equation numbers
After addressing the above comments the paper can be published in the journal.
The paper needs minor checking grammatically
Author Response
Dear respected reviewer,
First, we thank you for your consideration and helpful comments. The paper has been carefully revised in response to the comments raised. We feel that this helped to improve the quality of the manuscript. Please find enclosed a revised version of our manuscript for further consideration for publication. A full response to each comment is provided below, while key changes are also highlighted using blue text within the manuscript.
According to what the respected reviewer recommended, and consequently, after revising the manuscript based on the recommendations, the authors now hope that the respected reviewer finds the paper suitable for publication in the respected Journal of Applied Sciences.
- Remove the typo mistakes.
Thanks to the respected reviewer for this comment. The paper was reviewed to correct the Typo mistakes.
- The author should add one more section of Nomenclature before the references section:
Following the respected reviewer’s comment, the following nomenclature list was provided and added to the revised version of the manuscript:
|
DEA |
Differential Evolution Algorithm |
|
PF |
Pareto-Front |
|
NP |
Non-deterministic Polynomial |
|
VNS |
Variable Neighborhood Search |
|
MODE |
Multi-objective Differential Evolution |
|
NSGAII |
Non-Dominated Sorting Genetic Algorithm |
|
MOO |
Multi-objective Optimization |
|
SGA |
Standard Genetic Algorithm |
|
EP |
Evolutionary Programming |
|
FA |
Firefly-based Algorithm |
|
MOGA |
Multi-objective Genetic Algorithm |
|
MOEP |
Multi-objective Evolutionary Programming |
|
HEFT |
Heterogeneous Earliest Finish Time |
|
CPGA |
Critical Path Genetic Algorithm |
|
MFA |
Mean Field Annealing |
|
BGA |
Bi-objective Genetic Algorithm |
|
VAEA |
Vector Angle-Based Evolutionary Algorithm |
|
RVEA |
Reference Vector guided Evolutionary Algorithm |
|
GE |
Gaussian Elimination Graph |
|
FA |
Firefly based Algorithm |
|
R+ |
Sum of all better ranks |
|
R- |
Sum of all worse ranks |
|
α |
Significance level |
|
Cj(s) |
The time that processor pj finishes execution |
|
MaxjCj(s) |
Completion time of last processor in schedule s |
|
v ( j,s ) |
All tasks assigned to processor pj |
|
aft(s) |
The summation of all completion times divided by |P| |
|
λj |
The failure rate of processor pj |
|
λmn |
The communication failure rate of processors pm and pn |
|
Sim |
mapping task i to processor pm |
|
Sjn |
mapping task j to processor pn |
|
AT[ti] |
Readiness of each task to begin execution |
|
FT[ti] |
The Completion time of task i |
|
#p |
The number of tasks |
|
#t |
The number of processors |
- The reference list should be according to the journal format
Following the respected reviewer's comment, the reference style was corrected.
- Align the equations and equation numbers
The equation numbers were corrected, and the modifications were done according to the respected reviewer's comment.
Reviewer 2 Report
Comment 1
Please point out the why DE needs to be improved, with considering the problem characteristics.
Comment 2
Please enhance the introduction to the existing metaheuristics for solving the multi-objective task graph scheduling problem, which should be the motivation of the proposed algorithm.
Comment 3
The space at the beginning of line 126 looks strange.
Comment 4
The format of the equation and the corresponding numbers should be corrected.
Comment 5
The related should be improved by classifying the current methods into different categories and presenting a deep discussion on them.
minor revision is needed on the language
Author Response
Dear respected reviewer,
First, we thank you for your consideration and helpful comments. The paper has been carefully revised in response to the comments raised. We feel that this helped to improve the quality of the manuscript. Please find enclosed a revised version of our manuscript for further consideration for publication. A full response to each comment is provided below, while key changes are also highlighted using blue text within the manuscript.
According to what the respected reviewer recommended, and consequently, after revising the manuscript based on the recommendations, the authors now hope that the respected reviewer finds the paper suitable for publication in the respected Journal of Applied Sciences.
- Please point out the why DE needs to be improved, with considering the problem characteristics.
The authors thank the respected reviewer for the helpful comment. Apart from the fact that DE is a straightforward optimization method, it is also robust and powerful. Like many other optimization methods, DE operates based on some parameters and several operators. The aim of optimization methods is to explore a high-quality PF in an acceptable time, preventing early convergence to avoid local optimal solutions. It is obvious that the quality of operators as well as the solution representation scheme affect the ability of DE to find better PF and speed up convergence. Therefore, there is a rich literature to improve the performance of DE [20]. Likewise, hybridizing DE with other methods is another way to add more power to DE in discovering better PFs [21–23]. In our proposed method to optimize the well-known scheduling problem, not only operator improvement but also hybridization are applied to have a robust hybrid system to deal with objective functions (makespan, reliability, and flow time).
- Please enhance the introduction to the existing metaheuristics for solving the multi-objective task graph scheduling problem, which should be the motivation of the proposed algorithm.
Thanks to the respected reviewer for this comment. The following paragraph was added to the Introduction section after Figure 1. A pretty wide range of developed algorithms may be found in the literature to solve the task graph scheduling problem, which indicates the great importance of the problem in engineering applications. The state-of-the-art algorithms in the literature mostly apply soft computing methods and metaheuristics for solving the problem [11–14]. For instance, SGA and EP methods were developed by authors in [14], as were MOGA and MOEP methods applied in [15]. Moreover, HEFT-NSGA, MFA, weighted sum MOEP, hybrid algorithms, and a multi-agent system were developed in [8, 16–18], respectively. It can be seen that in the majority of state-of-the-art literature, a simple metaheuristic or weighted-sum method has been used. State-of-the-art methods are listed in Section III. Likewise, Section III includes several new evolutionary algorithms proposed in recent literature.
- The space at the beginning of line 126 looks strange.
Thanks to the respected reviewer for this precise comment. Spacing was checked and corrected at that point and whole the manuscript.
- The format of the equation and the corresponding numbers should be corrected.
The equation numbers were corrected and the modifications were done according to the respected reviewer comment.
- The related should be improved by classifying the current methods into different categories and presenting a deep discussion on them.
The authors thank the respected reviewer for this recommendation. The following paragraph was added at the end of section 3.
Due to having a wide range of state-of-the-art works in literature, it is useful to categorize the developed methods in terms of algorithm type and the problem type they are solving. The first categorization can be carried out in terms of the problem type which are bi-objective or three-objective task graph scheduling problems. Likewise, the second organization is done based on the algorithm type which can be either single-objective or multi-objective optimization approaches. The algorithms can be also either improved version or hybrid type algorithms. Table I represents the categorization of state-of-the-art methods. Three recently proposed evolutionary algorithms are also considered in the table.
Table I. Classification of the current methods in the literature
|
Method title |
Single-objective Algorithm (Applying weighted-sum) |
Multi-objective Algorithm |
Bi-objective TGS |
Three-objective TSG |
Reference number |
|
SGA |
√ |
× |
√ |
× |
[15] |
|
EP |
√ |
× |
√ |
× |
[15] |
|
Hybrid GA |
√ |
× |
√ |
× |
[15] |
|
EP |
√ |
× |
× |
√ |
[12] |
|
GA |
√ |
× |
× |
√ |
[12] |
|
MOGA |
× |
√ |
√ |
√ |
[14,31] |
|
MOEP |
× |
√ |
√ |
√ |
[14,32] |
|
HEFT |
√ |
× |
√ |
√ |
[17,34] |
|
CPGA |
√ |
× |
√ |
√ |
[17,35] |
|
HEFT-NSGA |
× |
√ |
√ |
√ |
[17] |
|
MFA |
× |
√ |
× |
√ |
[11] |
|
FA |
√ |
× |
√ |
× |
[13] |
|
MGA |
√ |
× |
√ |
× |
[42] |
|
BGA |
√ |
× |
√ |
× |
[41] |
|
MOO+Local Search |
× |
√ |
√ |
× |
[15] |
|
Ensemble System |
× |
√ |
√ |
√ |
[8] |
|
NSGA-II-WA |
× |
√ |
√ |
√ |
[43] |
|
VAEA |
× |
√ |
√ |
√ |
[44] |
|
RVEA |
× |
√ |
√ |
√ |
[46] |
The single-objective optimization algorithms use the weighted-sum method to be able to optimize more than one objective [47–49]. The reported results show that most of the multi-objective optimization methods outperform the single-objective and weighted-sum approaches.
Reviewer 3 Report
See attached PDF file

That is also mentioned in the paper. There are too little articles "the" and "a(n)" and some other errors as well (if I remember right, there were inappropriate capitalizations in the text).
Author Response
Dear respected reviewer,
First, we thank you for your consideration and helpful comments. The paper has been carefully revised in response to the comments raised. We feel that this helped to improve the quality of the manuscript. Please find enclosed a revised version of our manuscript for further consideration for publication. A full response to each comment is provided below, while key changes are also highlighted using blue text within the manuscript.
According to what the respected reviewer recommended, and consequently, after revising the manuscript based on the recommendations, the authors now hope that the respected reviewer finds the paper suitable for publication in the respected Journal of Applied Sciences.
- Section 2 starts with describing the NSGAII method, a popular evolutionary method for multi-objective optimization. The NSGAII method is used in the comparison in Section 5, but it isn’t clear why this algorithm is explained at this point.
The authors thank the respected reviewer for this precise comment. The paragraph mentioned was unnecessary for Section 2. Therefore, it was removed from the section. Likewise, a brief explanation was added to the related studies section.
- Section 4 is devoted to the description of the new hybrid method proposed by the authors with the aid of a flow chart of the whole program and pseudocodes for the components. Unfortunately to me (not an expert in the field) the presentation is rather confusing and chaotic.
Following the respected reviewer's comment, this part, Section 4, was reviewed and revised to make it more understandable.
- Were the same conditions used for the new algorithm as in the old benchmarks?
As mentioned in Section 5, the values of all parameters regarding the MODE algorithm are given in Table I. In our Matlab® implementation, the generation size and processor number are adjusted according to the literature. It is worth mentioning that the VNS technique does not add much additional time complexity to the hybrid method because the small and fast version of VNS is applied.
- The lack of a high percentage of the necessary articles “the” and “a(n)” in English did not make it pleasant for me to read the paper.
The paper was reviewed and revised regarding this case.
- 3, l. 92: I don’t think that the abbreviation MOO (multi-objective optimization?) was explained earlier in this paper.
Thanks to the respected reviewer for this precise comment. As the related paragraph was about NSGA-II, it was unnecessary for that part, according to the first comment of the respected reviewer. Therefore, this part was removed from the manuscript.
- 3, l. 106: The same holds for PF (Pareto front?).
Following the respected reviewer’s hint, the Pareto-Front was added to the first appearing PF in the paper (Section 5).
- 3, Figure 2: What do the numbers beside the task names mean?
They are just unique names for different tasks. It also shows the number of tasks in the task graph. A brief description was added to the paper (Section 2).
- 4, up to the end of Section 2: I already mentioned that you should also write f1 , Cj (s), sim etc. in the text. Moreover, you should refer to equation (2) etc. instead of equation 2. Apart from these issues that I am only pointing out “globally”, I have some that I’ll point out in more detail.
The mentioned issues were solved throughout Section 2 for all equations and symbols.
- 4, l. 127: Apart from the fact that I’d rather call it scheduling completion time than scheduling complete time, the “s” between quotation marks doesn’t make any sense here and therefore should be deleted. The total scheduling completion time, also known as makespan, does not seem to have any other abbreviation than f1 in this paper and the letter s denotes the schedule, as described in l. 137. (IMHO, the letter s already should be explained a sentence earlier.)
The complete time was a writing mistake, and it was corrected in the revised version. In addition, "s" was removed from the text. The symbol s denotes the scheduling in the paper, and it points to the scheduling represented by the solution representation scheme in figure 3. The related sentence was added to the paper.
- 4: You use Cj (s) (capital letter) in eqs. (2) and (3) and lines 130 and 131 but cj (s) (small letter) in eqs. (4) to (6). According to l. 139, the completion times are still meant.
Thank the respected reviewer for this hint. It was corrected in the revised version of the manuscript.
- 4, eq. (3) and l. 135: I think that wij should not be the finish(ing) time/end time but the execution time as in Figure 6 in Section 4; otherwise it doesn’t make sense. Moreover, a task is called “vi” here and ti in Section 4.
Wij is the time processor pj finishes executing task vi. A sentence was added to the paper for more clarification. Also, vi in problem descriptions and ti in algorithm descriptions are the same. Vi means vertices in the task graph, and ti is the name of vertices assigned to tasks. A sentence was added to the paper for more clarification.
- 4, l. 134: I’d write “In equation (3), all tasks assigned to processor pj belong to a set denoted by v(j, s).”
Thank the respected reviewer for this hint. It was corrected in the revised version of the manuscript.
- 4, eq. (5): The superscript should be a j, not an i: Pj succ(s).
Thank the respected reviewer for this hint. It was corrected in the revised version of the manuscript.
- 4, eq. (7): First of all, I don’t understand what the argument V should be – IMHO the communication reliability should only depend on the schedule, which was called s before, not S. Moreover, I don’t think that the right-hand side should be a double exponential but it should probably simply read
The section was revised according to the respected reviewer’s comment.
- 4, l. 153: I think that the sentence should read “In equation (7), the rate of communication failure between the processors pm and pn . . . ”.
This sentence was revised according to the respected reviewer’s comment.
- 4, l. 154: Please decide whether you call the quantities sim and sjn as in the displayed formula or Sim and Sjm. Moreover, does “indicate that . . . ” mean that sim is 1 if task i has been scheduled to processor m and sik = 0 for k 6= m? If so, you should write that down. Moreover, the quantity cij should be explained.
Following the respected reviewer’s comment, capital S symbols were changed to small letters. Likewise, cij is the communication cost between tasks i and j. The related section was revised.
- 6ff, Section 4: As I already mentioned, this section is rather confusing to me. The first figure that the authors refer to is Fig. 5 on l. 241, but Figures 3 and 4 are displayed first and have lower numbers. I’d present the flowchart Figure 5 first and describe which components of it are described more thoroughly in which pseudocode. I’d not call the pseudocodes “figures” but maybe “algorithms”. Moreover, I don’t like the way that the lines are numbered in the pseudocodes (arabic numbers for the program itself and small Roman numbers separately in each loop) but it would be better to number the lines from 1 to the end (or not at all) because that is the only way that would enable one to refer to a specific line easily, not like “line iii in the second While loop” or “line iv in the third If statement”. Finally, in this section you should also use indices/subscripts in the text, for example, in the first paragraph on p. 7.
Referring to Fig. 5, Fig. 2 and Fig. 3 were transferred to the right position. We used the figures to show the algorithms, but when we discuss them in the text, we treat them as algorithms, not figures. In addition, the numbering was removed from all algorithms. The subscripts issue was also solved.
- 7, l. 265: It’s rather processor pi than Pi Moreover, the quantities #p and #t were called |P| and |V | in Section 3 and in Figure 4. Please check the other notation in this paragraph as well whether it is consistent with Section 3.
The mentioned inconsistency was solved by changing # to ||.
- 7, l. 266: It’s “processors and tasks” (in this order).
This issue was corrected in the revised version of the manuscript.
- 7, Figure 5: In the step “Initialize the population . . . ” the archive is probably also initialized as the empty set. Moreover, please check the big rectangle in the middle. First, all dominance ranks seem to be replaced by their difference to the maximal one. Afterwards, three of them seem to be selected and it is not clear to me what is performed here, in particular, why the loop finishes before k exceeds PopSize. The line numbers here are more bothersome than useful. Finally, Random(0,Sum) means a uniformly distributed integer in the set {0, 1, . . . , Sum}, right?
Following the respected reviewer’s comment, archive initialization was also added to the first step. The k variable is used to find the index of the selected solution to be used for accessing the solution itself. The while also finishes when the solution is found. Likewise, the line numbers were removed for more clarity. Meanwhile, Random(0, Sum) is a random integer number between 0 and Sum.
- 8, Figure 6: I’d call that algorithm “computation of the makespan for a given schedule”. The “i=0” at the end of line 1 should be deleted. The other assignments of that line are a very clumsy way to say that three vectors are initialized as zero vectors. In the second comment to that line, I’d write “P[pi] is the time at which processor pi becomes idle”. Line ii in the outer for loop should be deleted – AT[ti] was initialized to 0 at the beginning, but it might have been modified in the inner for loop. In line ii of the inner for loop you probably mean “If . . . is not the same . . . ”.
The title of Fig. 6 was changed, and i = 0 was removed. The comment was also revised. AT [ti] was also deleted. Likewise, "If... is the same... " was changed to "If... is not the same... ".
- 8, l. 282: I’d not call Fig. 7 a flowchart but a pseudocode. Fig. 5 is the only flowchart in this paper.
Considering the respected reviewer’s hint, the necessary modifications were made in the revised version of the manuscript.
- 8, Figure 7: In line i of the inner for loop, something seems to be cut off, maybe “ra” should read “randomly”? And maybe the pseudocodes are called “figures” because they are scans from an earlier paper?
The issue related to "ra" was solved; it was about the wrong crop operation.
- 9, Figure 8: random(0,1) means a uniformly distributed random number in the interval (0, 1), right? And RandomNumber(1,n) seems to be the same as Random in Figure 5 (see above). Is Cutpoint1 smaller than Cutpoint2? Their assignments do not indicate that. Please explain.
Thanks to the respected reviewer for this comment. Yes, it is a random float number between 0 and 1. Cutpoint 1 should be smaller than Cutpoint 2. More description was added to the paper.
- 10, Figure 9: The first quantity that is initialized in line 5 should probably be p, not P. Line 6 contains a variable called cutpoint – is that an input parameter of the program or is a line assigning it missing?
P was changed to p. The cutpoint is a random number between 0 and n. A necessary description was added to the algorithm.
- 10ff, Section 5: In order to understand that section better, I numbered the test results on the margin. That should not be the task of the reader but the author should present the results in a clearer way, for example, by dividing the section into the subsections 5.1 to 5.5, or by preceding each result with Example/Experiment XXX. The examples 2 and 3 should also at least be described in the text as the first one, not only as “benchmarks” (each seems to be only a single benchmark). Moreover, I don’t think it is a good style to use figures in very different formats, probably taken from previous papers. In three of the five examples, the results with other algorithms are taken from papers where one of the authors is a coauthor, so it might be easier to guarantee that the conditions of execution were the same, but how about the other two examples? Why are the competitors against the new algorithm not the same in all examples?
The authors thank the respected reviewer for this comment. Four subsections, such as parameter values, objective function values, statistical experiments, and comparisons with the benchmarks, were defined as the subsections of Section 5. Moreover, additional explanations were added to each subsection.
- 11, l. 345: I’d rather write “To evaluate the performance . . . ”.
This sentence was revised according to the respected reviewer’s comment.
- 11, l. 357: Correct “cosidered” to “considered”.
The authors apologize for this mistake. It was corrected in the revised version of the manuscript.
- 12, Figures 11 and 12: In Figure 12 and a bit also in Figure 11 seems to be cut off below, which seems to indicate that the figures are scans as well.
Thanks to the respected reviewer for this hint. The necessary modifications were made to have a better shape for Figures 11 and 12.
Reviewer 4 Report
Manuscript ID: applsci-2348199
Journal: Applied Sciences
Title: A New Hybrid Algorithm based on Improved MODE and PF 1 Neighborhood Search for Scheduling Task Graphs in Hetero-2 geneous Distributed Systems
Authors: Nasser Lotfi and Mazyar Ghadiri Nejad
The solution to the Multi-objective task graph scheduling is expected to satisfy all scheduling objectives. Pretty large state-of-the-art algorithms exist in the literature that mostly applied different metaheuristics for solving the problem. This study proposes a new hybrid algorithm com- prising improved multi-objective differential evolution algorithm (DEA) and Pareto-front neighborhood search to solve the problem. The proposed method improves the performance of differential evolution by applying appropriate solution representation and effective selection, crossover, and mutation operators. Likewise, the neighborhood search algorithm is applied to improve extracted Pareto-front and speed up the evolution process. The effectiveness and performance of the developed method are assessed over well-known test problems collected from the related literature. The results confirmed that the developed algorithm outperforms all proposed methods considering the performance and quality of objective values. The paper is well written, interested and the results are good, I would like to suggest the following MINOR corrections before acceptance:
ــــــــــــــــــــــــــــــــــــــــــــــــــــــــــــــــــــــــــــــــــــــــــــــــــــــــــــــــــــــــــــــــــــــــــــــــــــــــــــــــــــــــــــــــــــــــــ
(1) A professional proofreading revision is strongly required. Typos must be corrected.
(2) Please add more details about the studied model
(3) The introduction must be reformulated to contain literature and future works, the main aim of the work.
(4) The arrangement of the manuscript should be added in a paragraph at the end of the introduction.
(5) The authors should clearly state the advantages of the used technique and a summary of the literature in the introduction.
(6) The results are interesting, also there was a great effort done, but there are NO clear applications of these results. If the authors can add some applications of their results this will be great
(7) I didn’t see any mathematical model for the problem under study, I think if there is a mathematical model that describes the studied problem it should be added or minimum referring to it in some references.
(8) Some new works on the studied problem should be added, this will improve the paper.
(9) The authors should revise and carefully arrange the references according to the guidelines of the journal.
Thanks a lot, to the editorial board of the Journal of Applied Sciences.

Author Response
Dear respected reviewer,
First, we thank you for your consideration and helpful comments. The paper has been carefully revised in response to the comments raised. We feel that this helped to improve the quality of the manuscript. Please find enclosed a revised version of our manuscript for further consideration for publication. A full response to each comment is provided below, while key changes are also highlighted using blue text within the manuscript.
According to what the respected reviewer recommended, and consequently, after revising the manuscript based on the recommendations, the authors now hope that the respected reviewer finds the paper suitable for publication in the respected Journal of Applied Sciences.
- A professional proofreading revision is strongly required. Typos must be corrected.
Thanks to the respected reviewer for this comment. The paper was reviewed to correct the typographical mistakes.
- Please add more details about the studied model
The authors appreciate the respected reviewer's comment. More details about the proposed model were added to Section 4.
- The introduction must be reformulated to contain literature and future works, the main aim of the work.
Following the respected reviewer's comment, the introduction section was rechecked, and a separate paragraph was added to the end of this section containing more literature, future studies, etc.
- The arrangement of the manuscript should be added in a paragraph at the end of the introduction.
Considering the respected reviewer's recommendation, the necessary paragraph was added to the end of the Introduction section.
- The authors should clearly state the advantages of the used technique and a summary of the literature in the introduction.
Thanks to the respected reviewer for this hint. This section was modified according to the respected reviewer's request.
- The results are interesting, also there was a great effort done, but there are NO clear applications of these results. If the authors can add some applications of their results this will be great.
The authors thank the respected reviewer for this helpful comment. Section 4 of the manuscript was revised considering this comment, and more descriptions were added in this regard.
- I didn’t see any mathematical model for the problem under study, I think if there is a mathematical model that describes the studied problem it should be added or minimum referring to it in some references.
With respect to the reviewer, the mathematical model of the problem and equations related to the objectives are given in Section 2. The formulas are represented by equations 1 to 8.
- Some new works on the studied problem should be added, this will improve the paper.
Thanks to the respected reviewer for this suggestion. The most recent works are given in Section 3. Moreover, to be more specific, the classification of the given methods was added to Section 3 as Table I.
- The authors should revise and carefully arrange the references according to the guidelines of the journal.
The authors appreciate the respected reviewer's comment. The reference type was modified considering the author guidelines of the journal and checking some recent published articles by the journal.
Reviewer 5 Report
This paper investigates the application of a hybrid algorithm for specific problem. There are some major issues so I need to Consider the paper again after revision:
1- Literature review is poor.
2- Novelty of this work is not stated properly. It is quite unclear what the novelty of this paper are. In this regard, a complete discussion should be prepared in the introduction section about the novelty and the main concerns of this paper. In which areas the novelty can be highlighted?
3- Regarding the fact that this paper is considered by authors as a paper in optimization field, it is not clear how the optimization procedure is performed and what is the complexity of the proposed hybrid technique. This point should be clarified properly.
4- Why GA and no other recent powerful techniques for conducting comparative investigations? I am sure that utilizing the GA as partially a classic algorithm can be of any justifications which is not enough.
5- Too many references are cited. What are the differences of your work with others?
16- The optimization problem statement in this paper is not in an acceptable level for a journal paper. A new section for this purpose is required.
27- The conclusion and the Abstract of the paper lacks the main quantitative results of this manuscript. It should be noted that some numerical results of the paper should be mentioned properly in the conclusion section.
38- Number of the Objective Function Evaluation (OFE) in the utilized methods and the modified version alongside the other methods in the revision should be provided in separate table for comparative purposes.
Author Response
Dear respected reviewer,
First, we thank you for your consideration and helpful comments. The paper has been carefully revised in response to the comments raised. We feel that this helped to improve the quality of the manuscript. Please find enclosed a revised version of our manuscript for further consideration for publication. A full response to each comment is provided below, while key changes are also highlighted using blue text within the manuscript.
According to what the respected reviewer recommended, and consequently, after revising the manuscript based on the recommendations, the authors now hope that the respected reviewer finds the paper suitable for publication in the respected Journal of Applied Sciences.
- Literature review is poor.
Thanks to the respected reviewer for this comment. The most recent works are added in Section 3. Moreover, to be more specific, the classification of the given methods was added to the literature review section as Table I.
2- Novelty of this work is not stated properly. It is quite unclear what the novelty of this paper are. In this regard, a complete discussion should be prepared in the introduction section about the novelty and the main concerns of this paper. In which areas the novelty can be highlighted?
The authors appreciate the respected reviewer's recommendation. The introduction section was improved to cover the missing information mentioned.
3- Regarding the fact that this paper is considered by authors as a paper in optimization field, it is not clear how the optimization procedure is performed and what is the complexity of the proposed hybrid technique. This point should be clarified properly.
Thanks to the respected reviewer for these points. The proposed optimization method has been deeply explained throughout the paper, especially in Section 4. However, more descriptions were added to the related sections to explain the optimization process more precisely. Population size and generation size are two effective parameters influencing time complexity. The values of these parameters are set similarly to the state-of-the-art methods to make the comparison fair. As can be seen in Table II, the population size and generation size are set to 200 and 300, respectively. Therefore, the time complexity of the proposed method is similar to that of its competitors. An extra explanation was added to Section 5.
4- Why GA and no other recent powerful techniques for conducting comparative investigations? I am sure that utilizing the GA as partially a classic algorithm can be of any justifications which is not enough.
The authors appreciate the respected reviewer's recommendation. According to Section 5, apart from GA (MOGA and NSGA), the comparison was done against many different new methods like EP, hybrid GA, MOEP, HEFT, HEFT-NSGA, MFA, FA, MGA, MOO+Local Search, Ensemble System, NSGA-II-WA, VAEA, and RVEA.
5- Too many references are cited. What are the differences of your work with others?
Thanks to the respected reviewer for these hints. It is noted that the number of references was 38, and considering different methods, algorithms, and their combinations as the proposed solution method, it is not possible to decrease it. In addition, the difference between our work and others is that we hybridized DE with VNS to explore the search space better and extract more promising solutions. We also improved DE by enhancing the operators as well as solution representation. Meanwhile, as it is mentioned in the paper, some of the methods have applied a single-objective optimization algorithm with a weighted-sum approach, but our method is a multi-objective algorithm.
6- The optimization problem statement in this paper is not in an acceptable level for a journal paper. A new section for this purpose is required.
Following the respected reviewer's suggestion, Section 2 has been reserved for the problem statement, which describes the problem, modeling, and equations in detail. However, more descriptions were added to the section to make it more specific.
7- The conclusion and the Abstract of the paper lacks the main quantitative results of this manuscript. It should be noted that some numerical results of the paper should be mentioned properly in the conclusion section.
The authors thank the respected reviewer for these recommendations. The obtained results for the main variables of the study, such as time, reliability, and flow time, were added to both sections of Abstract and Result.
8- Number of the Objective Function Evaluation (OFE) in the utilized methods and the modified version alongside the other methods in the revision should be provided in separate table for comparative purposes.
Thanks to the precise comment of the respected reviewer. Following the suggestions, the related table was added to Section 3 as Table I.
Reviewer 6 Report
The paper proposed a new hybrid algorithm for multi-objective task graph scheduling problem. Generally, the paper presented a good work, detailed each part of the designed algorithm and conducted extensive experiments. In my view, some aspects can be improved.
1. The presentation of the first two sections can be improved.
(1) There is much description about the work in Introduction, while the motivation is relatively weak.
(2) The literature review is presented mainly from the perspectives of problem and solution methods, while this two parts are organized in an unclear way. For the reference about solution approach, perhaps a comparison table can be given to increase the readability and more clearly presentation the contribution of this work.
2. There are much computational results. It can be presented in a more logical way. I suggested firstly give an overall description of the experiments about aims, kind of experiments, …, and then given each part in subsections.
Generally, the English writting is good and is easy to understand.
Author Response
Dear respected reviewer,
First, we thank you for your consideration and helpful comments. The paper has been carefully revised in response to the comments raised. We feel that this helped to improve the quality of the manuscript. Please find enclosed a revised version of our manuscript for further consideration for publication. A full response to each comment is provided below, while key changes are also highlighted using blue text within the manuscript.
According to what the respected reviewer recommended, and consequently, after revising the manuscript based on the recommendations, the authors now hope that the respected reviewer finds the paper suitable for publication in the respected Journal of Applied Sciences.
- The presentation of the first two sections can be improved.
Thanks to the respected reviewer for this comment. The content of both sections was expanded following this comment.
- There is much description about the work in Introduction, while the motivation is relatively weak.
The authors appreciate the respected reviewer's helpful point. The content of the Introduction section was improved to cover the missing information in the revised version of the manuscript.
- The literature review is presented mainly from the perspectives of problem and solution methods, while this two parts are organized in an unclear way. For the reference about solution approach, perhaps a comparison table can be given to increase the readability and more clearly presentation the contribution of this work.
Thanks to the respected reviewer for this constructive recommendation. Following this recommendation, an extra description was added to the manuscript. In addition, a classification and comparison table were added to Section 3 to clear up this issue.
- There are much computational results. It can be presented in a more logical way. I suggested firstly give an overall description of the experiments about aims, kind of experiments, …, and then given each part in subsections.
The authors thank the respected reviewer for this comment. Four subsections, such as parameter values, objective function values, statistical experiments, and comparisons with the benchmarks, were defined as the subsections of Section 5. Moreover, additional explanations were added to each subsection.
Round 2
Reviewer 3 Report
See attached PDF file

It's the same problem as before (lack of articles the/a(n)), but only in some parts like Section 2 and the beginning of Section 3, for example.
Author Response
Dear respected reviewer,
First, we thank you for your consideration and helpful comments. The paper has been carefully revised in response to the comments raised. We feel that this helped to improve the quality of the manuscript. Please find enclosed a revised version of our manuscript for further consideration for publication. A full response to each comment is provided below, while key changes are also highlighted within the manuscript.
- It's the same problem as before (lack of articles the/a(n)), but only in some parts like Section 2 and the beginning of Section 3, for example.
Thanks to the respected reviewer for this comment. The manuscript was rechecked and the articles were added to the text.
- The nomenclature list at the end of the paper still has some shortcomings. I think that it would be more convenient for the reader to sort all pure letter abbreviations (DEA to FA) alphabetically.
The nomenclature list was corrected according to the respected reviewer comment.
- I only noticed in this version that the abbreviations DEA and DE are used in the paper to refer to the same algorithm – please choose one (I think that DE is the abbreviation more commonly used).
The abbreviation DE was used in whole the text.
- The nomenclature list also shows that the text program used by the authors seems to have a problem with displaying indices – I’d rather write pj than pj.
The nomenclature list was corrected considering this comment.
- Another issue is the division of Section 5 into subsections – it is not what I meant and makes this section even more confusing. What I meant is that I had the impression that you were using different sets of test problems (or benchmarks), some containing only two objective functions, some of them three, and you should indicate somehow when you start discussing one set of test problems and when you start another. Subsection 5.1 is clear, it contains the parameters for the MODE algorithm, but that’s only the part up to Table II.
The remaining part after Table II was moved to the beginning of section 5.2.
- Subsection 5.2 has a quite insignificant title and seems to contain the first test problem(s) or benchmark and the part of Subsection 5.1 after Table II would also belong to it.
The title of section 5.2 was changed to “Performance evaluation using bi-objective benchmarks”.
- The title of Subsection 5.4 “Comparison with the benchmarks” is also a bit strange when you seem to have discussed two benchmarks already and this subsection seems to discuss three more, one starting at the beginning, one on p. 17 and the last one on p. 18.
The title of section 5.3 was changed to “Performance evaluation using three-objective benchmarks”. Also, Section 5.4 was removed.
- The titles of the subsections suggest that you discuss different aspects of the same benchmark, when I would have liked to have sections that indicate when “benchmark 1” is discussed and when the discussion of “benchmark 2” starts etc.
The related section was modified according to the reviewer’s request.
- In some parts of the paper the English seems to have been revised carefully, but some other parts, like Section 2, still contain too little articles. So I’d still advise to have a thorough language check done by the publisher.
The language level of the manuscript was improved in this version.
- In the last paragraph of p.2, the abbreviation “VNS” is explained twice.
The redundant explanation was removed.
- I don’t understand the phrase “As future work shows” on l. 106 – future work can’t show anything because it has’t been done yet.
It was changed to “As future works”.
- To be precise, since we are already at the end of Section 1, there are only five remaining sections (2 to 6) on l. 111.
It was changed to five.
- On l. 135, I’d rather optimize all objective than satisfy them.
All were changed to “optimize”.
- In the formulas (5) and (6) there is still c when it should be C.
They were changed to C.
- On l. 169 pm should be written with an index like pn.
The related parameter was modified according to the respected reviewer’s comment.
- The abbreviation GA, used in Section 3, should also be added to the nomenclature list.
The nomenclature list was modified according to the respected reviewer’s suggestion.
- The explanation before Figure 3 says that this figure assigns tasks ti to vertices vj, but to me it seems to assign tasks ti to processors pj
The name of the parameter in this section was corrected.
- In the paragraph below Figure 4 on p. 8, you again avoid indices, i.e. you write pi instead of pi Moreover, the number of processors and tasks on l. 317 should be |P| and |T|, not just P and T.
The indices were corrected in the revised version of the manuscript considering the respected reviewer’s comment.
- The caption of Figure 5 should read “Proposed Hybrid Method”, not “Hyprid Proposed Method”.
The caption of Figure 5 was modified in the revised version of the manuscript.
- Starting on p. 15, you first have Table IV, then twice Table VI, Table VIII and Table IX –please check the numbering and the references to the tables.
The numbering of the tables was rechecked and corrected.
Reviewer 5 Report
The authors replied to my comments properly.
Author Response
Dear respected reviewer,
The authors thank the respected reviewer for the positive comment on the submitted manuscript.
Regards,